# Effect of T1 Slope on Disappearance of Cervical Lordosis after Posterior Cervical Double-Door Laminoplasty Based on Medical Informatics

**DOI:** 10.3390/brainsci13081189

**Published:** 2023-08-11

**Authors:** Yulin Zhao, Binglei Zhang, Baisheng Yuan

**Affiliations:** Department of Orthopedics, Qilu Hospital (Qingdao), Cheeloo College of Medicine, Shandong University, No. 758 Hefei Road, Qingdao 266035, China; 2831023023@163.com (Y.Z.); 13605328311@126.com (B.Z.)

**Keywords:** cervical spondylosis, sagittal balance, double-door laminoplasty, lordosis, neck pain, T1 slope, image processing

## Abstract

Cervical sagittal balance plays a pivotal role in spine surgeries as it has a significant impact on the clinical outcomes in cervical spine surgery. Image processing techniques have significantly improved the accuracy and precision of cervical surgical techniques. This study aims to investigate the effects of T1 slope (T1s) on the disappearance of cervical lordosis after posterior cervical double-door laminoplasty using medical informatics and radiographic measures. To do so, we determined and measured the loss of T1s and cervical lordosis during the postoperative follow-up period in patients with double-door posterior cervical laminoplasty. Patients (n = 40) who underwent posterior cervical double-door laminoplasty participated in this study. For all patients, the difference between the preoperative T1s (angle between the upper edge of T1 and the horizontal line) and preoperative and postoperative cervical lordosis (Cobb method) was estimated, and the linear relationship between the two was statistically analyzed to observe the influence of preoperative T1s on postoperative cervical lordosis disappearance. The average preoperative T1s was 23.54°, and the average preoperative cervical lordosis angle was 8.50°. After 1–20 months of follow-up (mean = 9.53 months), the average postoperative cervical lordosis was 8.50°, and the average loss of cervical lordosis was 0.22°. Twenty cases had different degrees of lordosis angle loss after the operation, with an average loss of 9.31°. All patients were divided into groups A and B, according to a mean value of T1s = 23.54°, of which T1S > 23.54° was group A and T1s < 23.54 was group B. Cervical lordosis was quantified by the C2–C7 Cobb angle. The Cobb angle difference of cervical lordosis was measured before and after the operation, and its correlation with preoperative T1s was assessed. The preoperative Cobb angle and cervical curvature changes in the two groups were statistically compared, and the difference between the two groups was statistically significant (*p* < 0.05). The group with a T1s > 23.54° had greater loss of preoperative Cobb angle and cervical curvature. In group A, the mean preoperative cervical disability index (NDI) was 32.4 ± 3.4, and the mean postoperative NDI score was 16.5 ± 2.1. The mean preoperative VAS scores of neck pain and neck pain were 5.41 ± 1.1 and 5.55 ± 0.3, respectively, and the improvement in neck pain was −0.2%. The mean preoperative NDI in group B was 30.1 ± 2.9, and the mean postoperative NDI score was 11.5 ± 3.1. The mean VAS score for preoperative neck pain was 5.11 ± 1.2, that for postoperative neck pain was 4.18 ± 0.7, and that for neck pain improved by 18%. There was a significant difference between the two groups (*p* < 0.05). The disappearance of cervical lordosis after posterior cervical double-door laminoplasty is an important cause of postoperative cervical spine pain. The T1s is meaningful for predicting the loss of postoperative curvature in patients undergoing posterior cervical double-door laminoplasty. This is especially true for patients with good preoperative cervical curvature without ankylosis and kyphosis but with a wide T1s.

## 1. Introduction

Cervical spondylotic myelopathy (CSM) is a degenerative disease with occult onset and progressive aggravation that leads to spinal cord dysfunction endangering human health [1,2]. Main risk factors of CSM are cervical disk herniation, cervical bone hyperplasia, ligament hypertrophy, or cervical canal stenosis [2,3,4]. For patients with multilevel CSM who do not respond to conservative treatment, surgery is the standard treatment. Cervical laminoplasty is a decompression procedure without fusion conducted to enlarge the cervical spinal canal via the posterior approach for treatment of CSM. Posterior cervical laminoplasty procedure is performed by two main techniques, including posterior cervical single-door laminoplasty and posterior cervical double-door laminoplasty [5,6,7].

Cervical sagittal balance plays a pivotal role in spine surgeries, particularly cervical laminoplasty due to its significant impact on the clinical outcomes in cervical spine surgery [8,9]. Moreover, cervical balance serves an important function in defining surgical strategy and predicting clinical outcomes following surgery [10,11,12]. In line with advances in image processing and modeling, different radiologic and imaging factors have been proposed to manifest cervical balance and to evaluate its usefulness [13]. Studies on cervical sagittal alignment were initially focused on normative data but then expanded into correlation with global sagittal balance, prognosis of various conditions, surgery outcomes, definition and classification of cervical deformity, and prediction of targets for ideal cervical reconstruction [13,14,15]. In recent years, different markers such as thoracic inlet angle (mimicking pelvic incidence in the thoracolumbar spine), T1 slope (T1s), and T1s minus cervical lordosis (TS-CL) have been introduced as pivotal parameters for assessment of cervical sagittal balance in cervical laminoplasty. Several studies have revealed the usefulness of these radiological parameters; however, the accurate measurement of T1s is a challenging procedure for many patients because of their short necks or impediment of shoulder shading. This fundamental limit impedes the usefulness and accuracy of these cervical parameters to all patients. 

Posterior cervical double-door laminoplasty is a common surgical method for the treatment of cervical degenerative diseases, such as posterior longitudinal ligament ossification, developmental spinal stenosis, and CSM, and has the advantages of safe operation, wide decompression range, and obvious expansion of spinal canal volume [16]. The theory is based on the expansion of the space behind the cervical spine, which moves the cervical spinal cord backwards, avoiding compression from the front. However, to pursue the effect of spinal cord retropulsion, surgical decompression inevitably causes the destruction of the stability of the cervical spine. Many patients have axial symptoms such as cervical kyphosis, neck, shoulder and back pain, and muscle spasm after surgery, which affect the efficacy of surgery. Kawaguchi first defined this pathology as axial symptoms in 1999 [17], and its incidence ranges from 5.2% to 61.5%. It is difficult to self-heal and seriously reduces the quality of life of patients after surgery [2,18]. Several studies have reported that the loss of physiological cervical curvature after posterior cervical surgery is an important cause of postoperative cervical axial pain [19,20,21,22].

The normal physiological curvature of the cervical spine is mild lordosis, which can increase the buffer and absorb the shock. Maintaining normal physiological curvature and reducing the loss of curvature are of great importance to maintain the long-term stability of the cervical spine and restore the biomechanical environment of the cervical spine. Maintaining normal physiological curvature and reducing curvature loss is of great significance for maintaining long-term stability of the cervical spine and restoring the biomechanical environment of the cervical spine. However, after posterior cervical surgery, due to the decreased stability of the cervical spine, normal physiological lordosis decreases and kyphosis is formed and the neck muscles, nu-ligament structure and joint capsule, and other tissue structures will be pulled to different degrees, which easily produce fatigue damage during cervical spine activity, resulting in axial pain. Current evidence shows that cervical kyphosis is a compensatory response to spinal cord compression and spinal canal volume reduction. Machino et al., in a large-scale analysis of radiographical parameters of patients with CSM, assessed changes in sagittal alignment and range of motion (ROM) after cervical laminoplasty [7]. They reported that 7.2% (33/457) of patients with physiological cervical lordosis before surgery developed kyphosis 3 years after surgery, while 63 patients with preoperative kyphosis underwent lordosis reconstruction after surgery [7]. Poor curvature can lead to changes in the biomechanical environment of each segment of the cervical spine, accelerating disc degeneration and cervical instability and even sequence abnormalities, aggravating neck symptoms [19]. This study aims to investigate the effects of T1 slope (T1s) on the disappearance of cervical lordosis after posterior cervical double-door laminoplasty using medical informatics and radiographic measures.

## 2. Materials and Methods

### 2.1. Study Design

All experimental procedures of this study were approved by the local ethics committee of Qilu Hospital, Cheeloo College of Medicine, Shandong University, Qingdao, Shandong, China (Ethics code: KYLL-KS-2022077) that completely coincide with the ethical standards and regulations of the studies on human beings set by the Helsinki declaration (2014). This was a clinical study conducted on patients (n = 65) with CSM or ossification of the posterior longitudinal ligament that underwent double-door posterior cervical laminoplasty in Department of Orthopedics, Qilu Hospital, Qingdao, Shandong, China, between January 2017 and December 2021. Patients who lost their intervals, had incomplete data, and had no preoperative or postoperative cervical disability index (NDI) or visual analog scale (VAS) scores were excluded. A total of 40 patients were enrolled, including 30 males and 10 females aged 38 to 85 years old, with an average age of 63.7 ± 1.45 years old. Operation grade: C3-7; medical history: 1–240 months, mean 35.7 months; follow-up time: 1–20 months, mean = 9.53 months. Persistent ossification of the posterior longitudinal ligament (OPLL) occurred in 20 cases, multilevel CSM occurred in 16 cases, and cervical spinal cord injury occurred in 4 cases. All patients in the study underwent posterior cervical double-door laminoplasty.

### 2.2. Surgical Procedures

Under general anesthesia, a midline incision was made after C2~C7, showing C2~C7 spinous processes and lamina. The C7 spinous process was cut and made into bone strips. From C3 to C7, the posterior median process was worn to the spinal canal. Then, the outer lamina was worn along the inner edge of the bilateral facet joint, keeping the inner plate of the vertebral plate. The dissected spinous and lamina were strutted posteromedial with spinous process distraction forceps. Open door widened spinal cord decompression. Then, the spinous process openings were opened on both sides, and hydroxyapatite spacers were fixed between the open spinous processes of C3–C7 by double 7-gauge sutures. The bone graft strip prepared at C7 was implanted at the hinge of the lamina on both sides.

### 2.3. Measurement Standard

Preoperative and postoperative cervical lordosis: neutral lateral radiographs of the cervical spine were taken before, after, and during the follow-up, and the cervical lordosis angle was measured by a computer (Cobb’s method: tangents are made from the lower edges of C2 and C7, respectively, and the angle between the tangents is used as the value of the cervical curvature) (Figure 1, Figure 2 and Figure 3). Differences in preoperative and postoperative cervical lordosis were compared as loss of lordosis. The cervical disability index (NDI) and visual analog scale (VAS) were calculated before and after surgery. All patients were divided into two groups, A and B, according to a mean value of T1s = 23.54°, where group A consisted of cases with T1S > 23.54° and group B consisted of cases with T1s < 23.54. The postoperative change in cervical lordosis was determined by comparing the preoperative and postoperative C2–C7 Cobb angles. Then, the difference in cervical lordosis was measured before and after the operation, and its correlation with preoperative T1s was assessed. The preoperative cervical lordosis and cervical curvature changes in the two groups were statistically compared.

### 2.4. Statistical Analysis

Statistical analyses of this study were performed with Statistical Package for Social Sciences (SPSS) (IBM SPSS Statistics Inc., Chicago, IL, USA, Windows version 19.0). The normal distribution of all the continuous variables was evaluated with Kolmogorov–Smirnov normality test. The variables that have normal distribution were presented as means ± standard deviation (SD) and variables with skewed distribution were presented in median (Inter Quartile Range, IQR). For the variables with no normal distribution, the nonparametric Kruskal–Wallis H test was used for comparative analyses of the variables between the different groups. Chi-square test was used for comparison of other enumeration variables. For all statistical analyses in this study, the statistically significant difference was set at *p* = 0.05.

## 3. Results

The average preoperative T1s was 23.54° and the average preoperative cervical lordosis angle was 8.50°. After 1–20 months of follow-up (mean = 9.53 months), the average postoperative cervical lordosis was 8.50° and the average loss of cervical lordosis was 0.22°. Twenty cases had different degrees of lordosis angle loss after the operation, with an average loss of 9.31°. According to the mean of 23.54°, all patients were divided into groups A and B, of which T1s > 23.54° was group A and T1S < 23.54° was group B (Table 1). 

The Cobb angle difference of cervical lordosis was measured before and after the operation, and its correlation with preoperative T1s was compared. The preoperative Cobb angle and cervical curvature changes in the two groups were statistically analyzed, and the difference between the two groups was statistically significant (*p* < 0.05). The group with a T1s > 23.54° had greater loss of preoperative Cobb angle and cervical curvature. In group A, the mean preoperative NDI was 32.4 ± 3.4 and the mean postoperative NDI score was 16.5 ± 2.1. The mean preoperative VAS scores of neck pain and neck pain were 5.41 ± 1.1 and 5.55 ± 0.3, respectively, indicating an impaired improvement of −0.2% in neck pain (Figure 4). In group B, the mean preoperative and postoperative NDI score was 30.1 ± 2.9 and 11.5 ± 3.1, respectively. The mean preoperative neck pain VAS was 5.11 ± 1.2, whereas the postoperative mean was 4.18 ± 0.7, indicating an improvement of 18% in neck pain (Figure 5). There was a significant difference between the two groups (*p* < 0.05) (Table 2).

## 4. Discussion

In recent years, in addition to cervical fracture and dislocation and spinal canal tumors, total laminectomy is still a common surgical method and has been gradually replaced by laminoplasty in the treatment of degenerative diseases of the cervix. Laminoplasty preserves the spinal canal structure of the posterior part of the cervical spine, thus maintaining the stability of the sagittal motion of the cervical spine to a large extent [16]. However, with the widespread use of laminoplasty, it is still found that the occurrence of cervical kyphosis cannot be completely avoided. The reason is that the posterior ligament complex (spinous process, supraspinal and interspinous ligament), which is used to maintain the static mechanical balance of the cervical spine, and the posterior muscles (cervical semispinous muscle, multifidus muscle, etc.), which are used to maintain the dynamic mechanical balance of the cervical spine, are damaged so that the stability of the cervical spine is still affected to different degrees after surgery [23].

Neck, shoulder, and back pain occur for a long time after laminoplasty, accompanied by acid swelling, stiffness, heaviness, and muscle spasm, which seriously affects the life and work of patients. Kawaguchi [17] defined this group of symptoms as axial symptoms in 1999. The cervical curvature was abnormal. When normal cervical lordosis is straightened or kyphotic, the posterior cervical muscles, posterior ligament complex, joint capsule, and articular process are abnormally stretched during cervical movement, resulting in fatigue-induced axial symptoms. Kawakami et al. [17] believed that the loss of cervical curvature after door opening was closely related to the degree of axial symptoms. Takeuchi et al. [24] reported that axial symptoms were related to kyphosis. 

## 5. Significance of Maintaining Normal Cervical Curvature

The average physiological lordosis of C2–C7 was 14.4°, and the weight-bearing axis in the sagittal plane was behind the C2–C7 vertebral body. Maintaining normal cervical lordosis in the sagittal plane can reduce the dependence on the posterior muscles of the cervical spine to maintain gravity balance. When the normal physiological lordosis of the cervical spine is small, kyphosis is formed, the neck muscles and ligaments and the deep tissue of the joint capsule are stretched, and neck activities can cause strain, neck and shoulder pain, soreness, and axial symptoms such as weakness. There may be obvious pressure points and muscle spasms during physical examination, which are related to the degree of cervical kyphosis. The more severe the kyphosis is, the more obvious the axial symptoms [25,26].

Normal cervical curvature can increase the buffer and absorb the shock, which is an important factor to maintain the physiological and motor functions of the cervical spine and spinal cord. Fang Wen et al. measured cervical X-ray films of 300 normal people and 300 patients with cervical spondylosis, and the normal C2–C7 cervical curvature was 20.2°. Abnormal curvature of the cervical spine is related to the biomechanical changes of the cervical spine, and the biomechanical changes caused by the abnormal curvature of the cervical spine can lead to aggravation of cervical spine degeneration [27]. As the T1s increases, so does C2–C7 lordosis to maintain forward gaze, resulting in a greater degree of lordosis curvature. Similarly, the C2–C7 SVA tended to increase with increasing T1s. It is difficult to maintain normal sagittal balance [8,12,28]. 

The cervical vertebral body usually takes approximately 36% of the axial load, and the rear complex provides 64% of the axial load and counteracts its stretch [29]. In the cervical vertebra rear structure, the nuchal ligament is very important to resist flexion and provide tension [30]. The cervical posterior ligament complex constitutes a tethered stretch structure of the posterior cervical spine. During flexion activity, most of the load on the cervical spine is balanced by ligamentous tissue. The integrity of the anterior and middle column support structure and the posterior stretch structure of the cervical spine is very important to maintain the cervical curvature [31].

Patients with traditional laminoplasty need to wear braces for approximately 3 months, which significantly reduces the postoperative patient motion range. The range of motion after 1 year was 53% preoperatively and even decreased to 35% after 7 years [32]. For patients with posterior longitudinal ligament ossification, the range of motion decreased more significantly. The longer the duration of wearing a neck brace, the more obvious the decline in mobility, the more severe the atrophy of the posterior neck muscles, and the greater the possibility of axial symptoms [23]. The posterior cervical ligament complex and the muscles attached to the posterior cervical extensor muscles, especially mainly the head and cervical hemispine muscles, are the main factors that maintain the static stability of the cervical spine [33]. After double-door surgery, the cervical posterior extensor muscles atrophy, adhere, and lose toughness and it is difficult to maintain the physiological curvature of the cervical spine, which leads to the anterior shift of the sagittal balance of the cervical spine, resulting in the loss of physiological lordosis.

The T1s plays an important role in assessing the sagittal balance of the cervical spine and the overall curvature of the cervical spine. Attention has been given to the importance of the T1s but no consensus has been reached on the parameters to predict cervical kyphosis after laminoplasty. Matz et al. suggested that the change in cervical curvature is caused by posterior structural damage after posterior cervical surgery. During the 2-year follow-up after surgery, the cervical curvature change in patients with a high T1s was greater than that of patients with a low T1s. They also reported that patients with a high T1s were more likely to have changes in spinal curvature and had twice the incidence of postoperative kyphosis [3]. Joe et al. believed that the balance of the sagittal plane of the cervical spine after cervical laminoplasty has a compensatory effect and the change in the sagittal plane of the cervical spine after surgery is related to the preoperative T1s [34]. However, some scholars believe that the change in the cervical sagittal plane after laminoplasty is not related to the T1s [3]. It has been reported that patients with a high preoperative T1s have a greater degree of postoperative cervical lordosis loss than patients with a low T1s. Such results have been previously reported in the literature [1,3,34] and it is believed that patients with a high T1 slope need more energy expenditure for the compensation of sagittal balance.

In groups with a larger T1S, greater effort and muscle may be required to maintain sagittal balance in the posterior aspect compared to groups with a lower T1S, and this effort is easily decompensated by disrupting the structure of the posterior neck [32,35]. When the Cobb angle of the cervical spine is negative, the erector spinae pull produces flexion, which further aggravates kyphosis. The changes in cervical curvature, straighter cervical spine, and kyphosis will accelerate the degeneration of the cervical spine, forming a vicious cycle [35].

This study aimed to find a preoperative method to evaluate the factors of loss of lordosis after surgery, predict the possibility of loss of lordosis after surgery, and judge the prognosis of surgery. The researchers selected T1s as a preoperative evaluation factor, and the results of this study showed that clinical parameters and health-related quality of life index showed that T1s was significant in predicting postoperative curvature loss in patients undergoing posterior cervical double-door laminoplasty. 

It should be noted that the double-door laminoplasty is relatively an old technique; however, in low- and middle-income countries, it is still a very important part of surgeons’ armamentarium as a low-cost and yet effective surgery. Alternative surgical approaches, including stabilization surgery with screws, with less cervical lordosis loss have been developed and used in some clinical practice. Although use of screws for stabilization surgery can reduce the angle of loss of cervical kyphosis, posterior double-door cervical laminoplasty is still a common method of operation in our institution. Internal fixation stabilization is performed only for patients with kyphosis (kyphosis less than 0°) or with cervical instability. Findings of this study improved our understanding of this issue and more and more patients with smaller kyphosis would be treated with internal fixation. 

The present study had some limitations that should be considered in interpreting the results and generalizing the findings to the general population. The first limitation was it was a small sample size and single-center study with limited data and a small number of cases. Second, this study lacked long-term follow-up data with in-depth observations. The third limitation of the study was it assessed only one independent variable, and other independent variables were not considered, including case type, operation time, and blood loss, so longer follow-up and multiple regression analysis are needed.

## 6. Conclusions

The disappearance of cervical lordosis after posterior cervical double-door laminoplasty is an important cause of postoperative cervical axial pain. The smaller T1s has better improvement of postoperative dysfunction index pain score and less loss of cervical curvature. The T1s is meaningful for predicting the loss of postoperative curvature in patients undergoing posterior cervical double-door laminoplasty. This is especially true for patients with good preoperative cervical curvature without ankylosis and kyphosis but with a large T1s.

## Figures and Tables

**Figure 1 brainsci-13-01189-f001:**
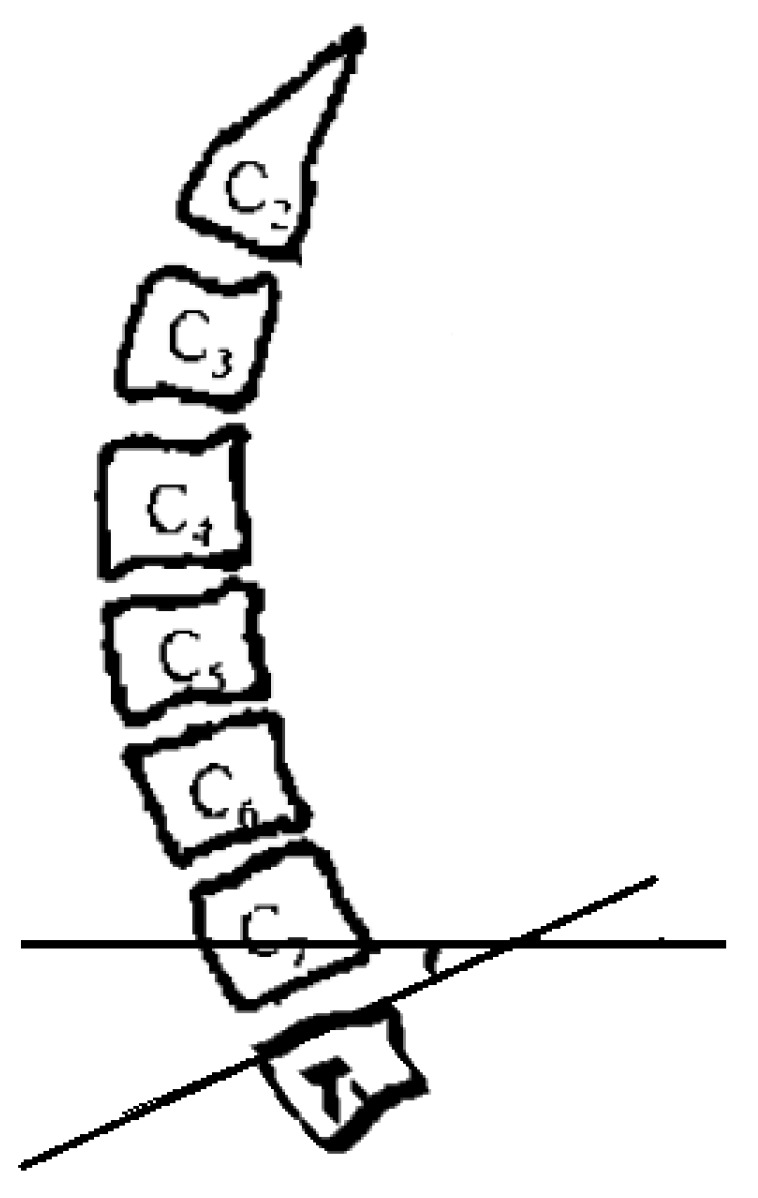
Preoperative T1s determination method used in the study. The angle between the upper edge of T1 and the horizontal line was measured.

**Figure 2 brainsci-13-01189-f002:**
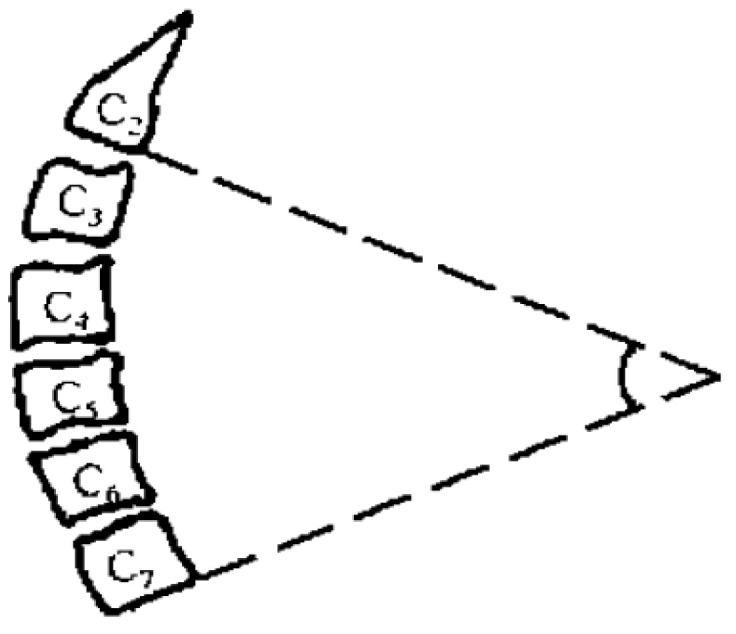
Cobb angle of cervical curvature. The angle formed by the parallel line of C2 lower endplate and C7 lower endplate represents the curvature of cervical vertebra.

**Figure 3 brainsci-13-01189-f003:**
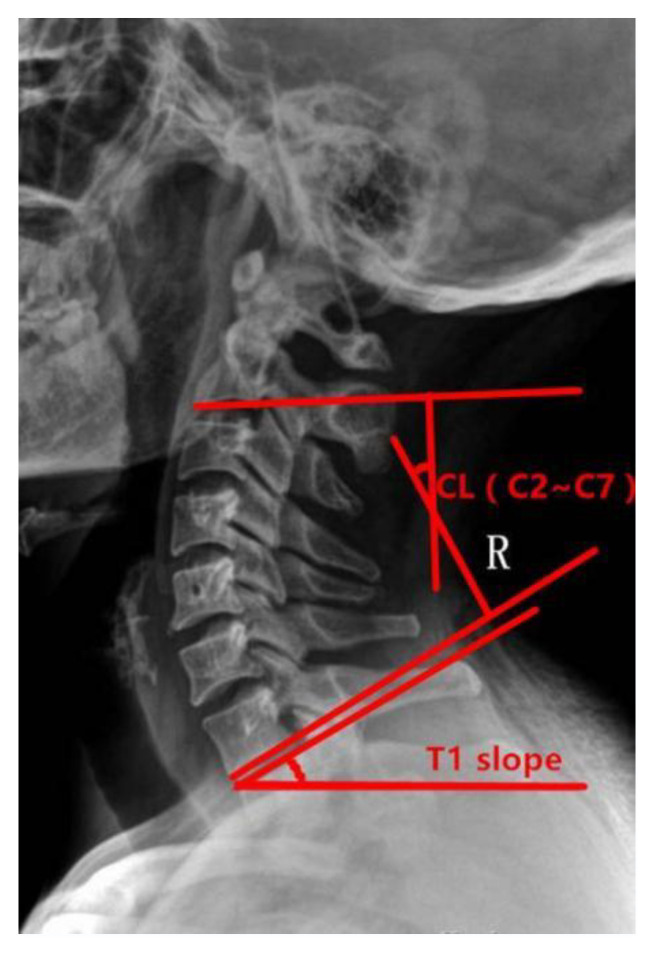
The cervical curvature and T1 inclination angle were measured by lateral radiograph of cervical spine. The measurement method is the same described in Figure 1 and Figure 2.

**Figure 4 brainsci-13-01189-f004:**
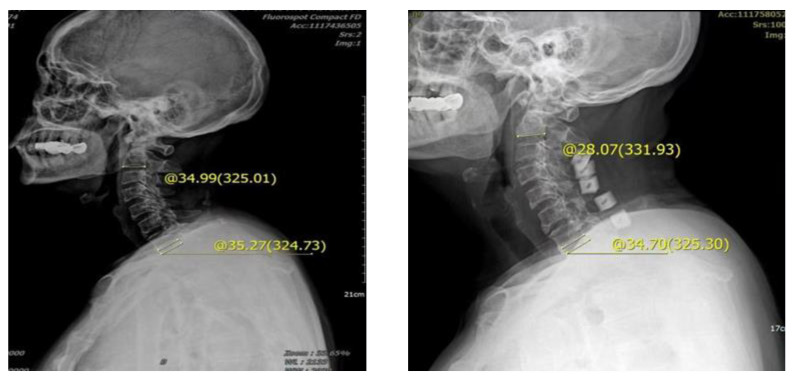
Male, 62 years old, preoperative diagnosis: CSM, cervical vertebra 3–7 posterior double door. The preoperative tilt angle of T1 was 35.27°, the preoperative cervical curvature was 34.99°, the postoperative follow-up was 3 months, the cervical curvature was 28.07°, and the cervical curvature lost 6.92°.

**Figure 5 brainsci-13-01189-f005:**
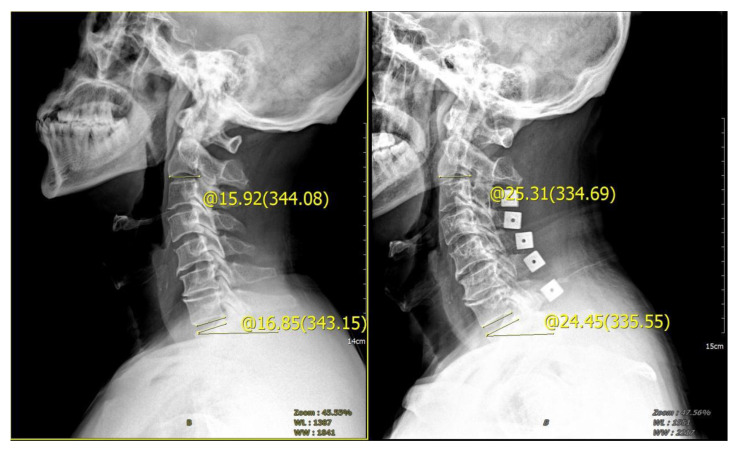
Male, 65 years old, diagnosed with CSM, cervical vertebra 3–7 posterior double door. The preoperative tilt angle of T1 was 16.85°, the preoperative cervical curvature was 15.92°, and the postoperative follow-up 3 months later showed that the cervical curvature was 25.31°, which increased by 9.39°.

**Table 1 brainsci-13-01189-t001:** Distribution of patients with or without cervical curvature loss after operation.

Curvature Loss	Case	Gender: M/F	T1s Preoperative
Curvature loss	20	14/6	23.94
Curvature no loss	20	16/4	23.15

**Table 2 brainsci-13-01189-t002:** Comparisons of the clinical parameters between two groups, A (T1s > 23.45°) and B (T1s < 23.45°), for preoperative and postoperative.

Group	Case	Average T1s (°)	Preoperative Cobb Angle of Cervical Spine (°)	Postoperative Cobb Angle of Cervical Spine (°)	Loss of Curvature (°)	Preoperative NDI	Postoperative NDI	Preoperative VAS	Postoperative VAS	VAS Improvement
Group A (T1s > 23.54°)	22	30.26 ± 9.95	10.84 ± 8.11	9.14 ± 5.33	1.71 ± 3.17	32.4 ± 3.4	16.5 ± 2.1	5.41 ± 1.1	5.55 ± 0.3	−0.2%
Group B (T1s < 23.54°)	18	15.34 ± 7.23	5.65 ± 5.09	8.23 ± 5.01	2.58 ± 3.11	30.1 ± 2.9	11.5 ± 3.1	5.11 ± 1.2	4.18 ± 0.7	18%

## Data Availability

The datasets used and analyzed during the current study are available from the corresponding author on reasonable request.

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
