# Peer review of "Effect of T1 Slope on Disappearance of Cervical Lordosis after Posterior Cervical Double-Door Laminoplasty Based on Medical Informatics"

_brainsci, 2023, doi:10.3390/brainsci13081189_

Round 1

Reviewer 1 Report

The current consensus of spine surgeons agrees that preoperative T1Slope is an important factor in postoperative kyphosis of cervical spine alignment, with greater T1Slope leading to less postoperative cervical kyphosis and poorer clinical score improvement.

I have the following questions regarding this study, and I would appreciate your answers

1. Are there any novelties found in the results of this study? Is it my understanding that you have confirmed previous findings?

2. About the method

1) Regarding the analysis of cervical spine alignment, C7 and T1 vertebrae are often difficult to confirm on Xp due to shoulder shadows.

(i) Regarding the left photograph in Fig. 4, is the cervical vertebrae in mid-position? Unlike Figure 5, it appears to be flexed backward. I have a doubt about the imaging method and the conditions under which the evaluation was made.

(ii) Did you use a CT sagittal section as shown in Fig. 4? If so, the C2-7 angle and T1Slope cannot be properly measured because they are different from the standing position under gravity, so the measurement method is still questionable.

(iii) Regarding NDI and VAS, can we treat patients with cervical cord injury in the case as we do with other diseases? There is a possibility that the findings may not improve after surgery due to the disease, so it may be necessary to exclude them or take other measures.

(iv) Regarding the follow-up period for patients, should we treat the situation of wound pain and other stiffness at 1 month postoperatively and 20 months in the same way? You stated that the mean was 9.53 months and the median 3.52 months. Would it affect the results if many early postoperative cases were included?

( v) CRegarding the comparison between groups A and B, why is it not mentioned in the Methods section instead of being implemented in the results? (Regarding Table 1, I did not understand whether you divided the patients into two groups, those with progressive kyphosis and those without kyphosis, or whether you divided them according to T1 slope above or below 23.54 degrees).

3. indications for surgery and extent of vertebroplasty

I did not understand the average C2-7 angle, but are there any restrictions at your institution, such as refraining from laminoplasty for patients who have always had kyphosis?

Is C3-7 the standard for laminoplasty? (I would like to confirm that C7 spinous process is often preserved these days to prevent kyphosis after surgery.)

4. Regarding the criteria for determining the size of the T1slope, in this study, the median (mean?) value is used. Is 23.54 degrees considered appropriate? I think it is also necessary for readers to know the appropriate index based on previous studies.

Author Response

Comments and Suggestions for Authors

The current consensus of spine surgeons agrees that preoperative T1Slope is an important factor in postoperative kyphosis of cervical spine alignment, with greater T1Slope leading to less postoperative cervical kyphosis and poorer clinical score improvement.

I have the following questions regarding this study, and I would appreciate your answers

 Answer: We thank you the respected reviewer for devoting time and efforts to review our manuscript and gave us the valuable comments. We carefully read all comments and accordingly revised the manuscript. We also provided our answer, on the one-by-one basis, below. We hope the revised manuscript satisfies the respected editorial team to consider it for publishing.

  1. Are there any novelties found in the results of this study? Is it my understanding that you have confirmed previous findings?

 Answer: In the clinical observation of this kind of phenomenon, after comprehensive review of the literature, we found that our findings are consistent with the findings of the previous studies indicating that our findings confirm the previous findings. Our innovation is that we took the average T1 inclination angle of 23.54 degrees as the cut-off point to study the imaging and surgical effect of postoperative cervical kyphosis in patients greater than and less than this value. This approach would have significant guidance value in clinical practice in future. With the increase of the number of samples in the future, this cut-off value will gradually approach a more accurate value.

  1. About the method

1) Regarding the analysis of cervical spine alignment, C7 and T1 vertebrae are often difficult to confirm on Xp due to shoulder shadows.

Answer: With regard to the analysis of cervical alignment, it is often difficult to confirm C7 and T1 vertebrae on Xp because of shoulder shadow.

(i) Regarding the left photograph in Fig. 4, is the cervical vertebrae in mid-position? Unlike Figure 5, it appears to be flexed backward. I have a doubt about the imaging method and the conditions under which the evaluation was made.

Answer: Is in a neutral position. It may be related to the patient's pain or neck and shoulder stiffness, and images that we consider not up to the criteria have been excluded from the group.

(ii) Did you use a CT sagittal section as shown in Fig. 4? If so, the C2-7 angle and T1Slope cannot be properly measured because they are different from the standing position under gravity, so the measurement method is still questionable.

Answer: That is a very good question and we appreciate it. The angle values of C2-7 and T1Slope are different from those of standing cervical vertebrae in positive and lateral position and recumbent position under gravity. We measured the X-ray before and after the operation. Figure 4 the postoperative X-ray of this patient, as mentioned in the above question, we need to adjust the contrast repeatedly, and when the T1Slope display is clear, the whole image is not very clear. Therefore, we replaced the image of another patient and showed it more clearly in the article.

(iii) Regarding NDI and VAS, can we treat patients with cervical cord injury in the case as we do with other diseases? There is a possibility that the findings may not improve after surgery due to the disease, so it may be necessary to exclude them or take other measures.

Answer: Thanks for the good point and comments. The main research aspects of the effect of T1 tilt angle on cervical kyphosis and surgical results, for spinal cord injury and other diseases, the same improvement of cervical kyphosis will also improve postoperative symptoms.

(iv) Regarding the follow-up period for patients, should we treat the situation of wound pain and other stiffness at 1 month postoperatively and 20 months in the same way? You stated that the mean was 9.53 months and the median 3.52 months. Would it affect the results if many early postoperative cases were included?

Answer: We are very sorry, regarding the report of follow-up figures, there were few mistakes in the manuscript that was due to translation error so that the patient's medical history of 35.6 months was mistakenly reported as the mean postoperative follow-up time (mean: 9.53 months). We are really sorry for it and the mistake has been corrected. The correct follow-up time should be an average of 9.53 months. During the follow-up period, we performed the same treatment for stiff conditions such as wound pain at 1 month and 20 months after operation. If more early postoperative cases are included, such as those more than 3 months old, the results will not be affected.

( v) Regarding the comparison between groups A and B, why is it not mentioned in the Methods section instead of being implemented in the results? (Regarding Table 1, I did not understand whether you divided the patients into two groups, those with progressive kyphosis and those without kyphosis, or whether you divided them according to T1 slope above or below 23.54 degrees).

 Answer: We added this section to the Methods section.

  1. indications for surgery and extent of vertebroplasty

I did not understand the average C2-7 angle, but are there any restrictions at your institution, such as refraining from laminoplasty for patients who have always had kyphosis?

Answer: Laminoplasty is not prohibited for patients with reduced kyphosis, and laminoplasty is avoided for patients with kyphosis (kyphosis less than 0 °) and cervical instability.

Is C3-7 the standard for laminoplasty? (I would like to confirm that C7 spinous process is often preserved these days to prevent kyphosis after surgery.)

Answer: C3-7 is not the standard of laminoplasty. We are preserving the C7 spinous process. The patients we included in the study were patients who underwent posterior cervical laminoplasty (C37). This is to unify the inclusion criteria.

  1. Regarding the criteria for determining the size of the T1slope, in this study, the median (mean?) value is used. Is 23.54 degrees considered appropriate? I think it is also necessary for readers to know the appropriate index based on previous studies.

Answer: There is no uniform standard, and we did not include the previous studies findings into our analysis. We designed a group with an average of 23.54 degrees to observe the difference in outcome of the operation and compared the effects between the larger group and the smaller group. The median was a typo and mean or average is correct. We changed all the representative word to mean.

Reviewer 2 Report

Authors present a retrospective study on 40 patients who underwent double door posterior laminoplasty in the cervical spine to estimate the effect of T1 slope on cervical lordosis. The group with a T1s >23.54° had greater loss of preoperative Cobb angle and cervical curvature. The disappearance of cervical lordosis after posterior cervical double-door laminoplasty, as an important cause of cervical pain, was brought into connection with  T1and its ability to prediction the loss of lordosis - greater the T1s, greater the loss of lordosis, greater the pain. Abstract is not very clearly written - it should be clearly stated which of the groups had poor outcome and which T1s value is predictive for poor outcome (lordosis loss, increased neck pain). Is this prospective or retrospective study? Major drawbacks are: low number of patients, different pathologies (CSM, OPLL), no control group (patients who underwent ventral surgery; patients who underwent stabilization etc). Double door laminoplasty, as described by the authors - is an outdated technique; however in low and middle income countries still a very important part of surgeons armamentarium as a low-cost surgery. I suggest to include into Discussion alternative surgical approaches to the pathology, especially from a standpoint that stabilization surgery with screws probably leads to less cervical lordosis loss. 

What is completely missing are illustrative cases and description of indications for surgery, therefore I suggest to include at least 3 cases with MRI pre and postoperative; CT. A table with list of all patients is needed for overview of the patient collective. Did all patients receive C3-7 surgery? 

Also one important issue : what is the clinical value? Would you treat patients with high T1s , or higher than 23, differently? Change of approach? Role of anterior surgery?

For Discussion include and comment:

Aflatooni J, Mohanty S, Angelov I, Hirase T, Bondar K, Kakareka M, Saucedo J, Casper D, Saifi C. Crossing the cervicothoracic junction: an evaluation of radiographic alignment, functional outcomes, and patient-reported outcomes. J Neurosurg Spine. 2023 Mar 3:1-9. doi: 10.3171/2023.1.SPINE221013. Epub ahead of print. PMID: 36883622. Niu Y, Lv Q, Gong C, Duan D, Zhou Z, Wu J. Predictive effect of cervical sagittal parameters and corresponding segmental paravertebral muscle degeneration on the occurrence of cervical kyphosis following cervical laminoplasty. World Neurosurg. 2023 Apr 7:S1878-8750(23)00486-2. doi: 10.1016/j.wneu.2023.04.011. E .      

Acceptable. 

Author Response

Comments and Suggestions for Authors

Authors present a retrospective study on 40 patients who underwent double door posterior laminoplasty in the cervical spine to estimate the effect of T1 slope on cervical lordosis. The group with a T1s >23.54° had greater loss of preoperative Cobb angle and cervical curvature. The disappearance of cervical lordosis after posterior cervical double-door laminoplasty, as an important cause of cervical pain, was brought into connection with  T1and its ability to prediction the loss of lordosis - greater the T1s, greater the loss of lordosis, greater the pain.

Answer: We thank you the respected reviewer for devoting time and efforts to review our manuscript and gave us the valuable comments. We carefully read all comments and accordingly revised the manuscript. We also provided our answer, on the one-by-one basis, below. We hope the revised manuscript satisfies the respected editorial team to consider it for publishing.

Abstract is not very clearly written - it should be clearly stated which of the groups had poor outcome and which T1s value is predictive for poor outcome (lordosis loss, increased neck pain).

Answer: We revised the abstract and edited some sentences to make it clear in methods and results of the study. 

Is this prospective or retrospective study?

 Answer: It is a retrospective study.

Major drawbacks are: low number of patients, different pathologies (CSM, OPLL), no control group (patients who underwent ventral surgery; patients who underwent stabilization, etc).

Answer: Thanks for great and valuables comments. Yes, the number of cases is small. However, considering the number of patients and the inclusion criteria, we could reach this sample size for this study. We are planning to conduct further studies with greater sample size on the ground of the findings of this study in the future. The purpose of our study was to study the effect of T1 tilt angle on the effect of posterior cervical laminoplasty, so there was no comparison between patients who underwent anterior surgery and those who received stable internal fixation. instead, the larger T1 tilt angle and the smaller T1 tilt angle were used as the control group to determine what kind of tilt angle had a greater impact on the effect of the operation.

Double door laminoplasty, as described by the authors - is an outdated technique; however in low and middle income countries still a very important part of surgeons armamentarium as a low-cost surgery. I suggest to include into Discussion alternative surgical approaches to the pathology, especially from a standpoint that stabilization surgery with screws probably leads to less cervical lordosis loss. 

Answer: we revised the discussion to include this topic. The use of screws for stabilization surgery can reduce the angle of loss of cervical kyphosis, but posterior double-door cervical laminoplasty is still a common method of operation in our institution. Internal fixation stabilization is performed only for patients with kyphosis (kyphosis less than 0 °) or with cervical instability. After our conclusion, we have a deeper understanding of this problem, and more and more patients with smaller kyphosis are treated with internal fixation. With the increase of cases, we will increase the control group with different surgical methods in the future study.

What is completely missing are illustrative cases and description of indications for surgery, therefore I suggest to include at least 3 cases with MRI pre and postoperative; CT. A table with list of all patients is needed for overview of the patient collective. Did all patients receive C3-7 surgery? 

 Answer: Routine preoperative magnetic resonance examination is performed in our institution, but postoperative follow-up does include magnetic resonance imaging, but cervical vertebra photography or CT. Surgical indications are multi-segmental (more than or equal to 3 segments) cervical Spondylotic myelopathy, cervical ossification of posterior longitudinal ligament, cervical spinal canal stenosis and other diseases. The patients we included in the study were C3-7 patients who underwent posterior cervical laminoplasty.

Also one important issue : what is the clinical value? Would you treat patients with high T1s , or higher than 23, differently? Change of approach? Role of anterior surgery?

 Answer:  It was a great question. In the past, we performed posterior double-door surgery on patients without kyphosis, with the progress of our study, in the future, we will improve kyphosis in patients with high T1 tilt angle (greater than 23.54 °) or in patients with loss of kyphosis after surgery combined with internal fixation in order to improve postoperative symptoms.

For Discussion include and comment:

 Answer: we included and discussed the mentioned references.

Editor tip: no need for authors answer

Aflatooni J, Mohanty S, Angelov I, Hirase T, Bondar K, Kakareka M, Saucedo J, Casper D, Saifi C. Crossing the cervicothoracic junction: an evaluation of radiographic alignment, functional outcomes, and patient-reported outcomes. J Neurosurg Spine. 2023 Mar 3:1-9. doi: 10.3171/2023.1.SPINE221013. Epub ahead of print. PMID: 36883622.

Niu Y, Lv Q, Gong C, Duan D, Zhou Z, Wu J. Predictive effect of cervical sagittal parameters and corresponding segmental paravertebral muscle degeneration on the occurrence of cervical kyphosis following cervical laminoplasty. World Neurosurg. 2023 Apr 7:S1878-8750(23)00486-2. doi: 10.1016/j.wneu.2023.04.011. E .      

Comments on the Quality of English Language

Acceptable. 

Answer: Thanks, for the comment. we asked an expert copy editor to revise the manuscript to address any language issues in the manuscript.

Reviewer 3 Report

General impression

In this study, the authors investigated the effects of T1 slope (T1s) on the disappearance of cervical lordosis after posterior cervical double-door laminoplasty using medical informatics and radiographic measures.   Through the results of current study, they conducted the facts that the disappearance of cervical lordosis after posterior cervical double-door laminoplasty is an important cause of postoperative cervical spine pain and the T1s is meaningful for predicting the loss of postoperative curvature in patients undergoing posterior cervical double-door laminoplasty. 

I think this paper includes interesting and valuable information for the spine surgeons.

The methodology of this study was precisely explained.  I could not find either error in writings or mistakes in the text.  However, I have a couple requests to be revised as stated below.  

*I ask the authors that correction parts will be shown in red color in the revised manuscript.

1. surgical procedure related to postoperative cervical kyphosis

  I want to know the surgical procedure with regard to care for muscles attached to C2 spinous process (rectus capitis posterior major, obliquus capitis inferior, semispinalis cervicis) and nuchal ligament.

2. in association with #1.

  I think one of the causes to occur postoperative cervical kyphosis by posterior approach must be crude treatment of soft tissues such as muscles attached to C2 spinous process and nuchal ligament.  When we perform posterior cervical surgery, we always make a great effort to keep muscle attachments of C2 spinous process and continuity of nuchal ligament (cut muscle fascia a little laterally either side).  I recommend the authors will add the comments about the ingenuity and possibility of surgical procedure to maintain the preoperative cervical alignment postoperatively in the discussion section.

Author Response

Answers to reviewer 3:

Comments and Suggestions for Authors

General impression

In this study, the authors investigated the effects of T1 slope (T1s) on the disappearance of cervical lordosis after posterior cervical double-door laminoplasty using medical informatics and radiographic measures.   Through the results of current study, they conducted the facts that the disappearance of cervical lordosis after posterior cervical double-door laminoplasty is an important cause of postoperative cervical spine pain and the T1s is meaningful for predicting the loss of postoperative curvature in patients undergoing posterior cervical double-door laminoplasty. 

I think this paper includes interesting and valuable information for the spine surgeons. The methodology of this study was precisely explained.  I could not find either error in writings or mistakes in the text.  However, I have a couple requests to be revised as stated below.  

Answer: we thank the respected reviewer for the time and efforts devoted to review our manuscript. We carefully review all comments and revised the manuscript accordingly. We also provided our answers to the comments below on the one-by-one basis. We hope the revised manuscript meets the editorial requirements of the journal to consider it for publishing in the journal.  

*I ask the authors that correction parts will be shown in red color in the revised manuscript.

Answer: we did the same and made all revisions in red highlighted.

  1. surgical procedure related to postoperative cervical kyphosis

  I want to know the surgical procedure with regard to care for muscles attached to C2 spinous process (rectus capitis posterior major, obliquus capitis inferior, semispinalis cervicis) and nuchal ligament.

Answer: Yes, our surgery will preserve the muscles attached to the C2 spinous process and cervical ligament as much as possible to prevent postoperative axial pain.

  1. in association with #1.

  I think one of the causes to occur postoperative cervical kyphosis by posterior approach must be crude treatment of soft tissues such as muscles attached to C2 spinous process and nuchal ligament.  When we perform posterior cervical surgery, we always make a great effort to keep muscle attachments of C2 spinous process and continuity of nuchal ligament (cut muscle fascia a little laterally either side).  I recommend the authors will add the comments about the ingenuity and possibility of surgical procedure to maintain the preoperative cervical alignment postoperatively in the discussion section.

Answer: As the respected reviewer mentioned, when we perform posterior cervical surgery, we always try to maintain the muscle attachment of the C2 spinous process and the continuity of the cervical ligament (the muscle fascia is slightly cut laterally on both sides). The C2 spinous process, supraspinous ligament, attached muscle and bilateral facet process were preserved in the operation, which played a certain role in maintaining the postoperative curvature. In recent years, it is considered that the occurrence of postoperative axial symptoms is related to many factors, such as segmental cervical movement, injury of posterior branch of spinal nerve, destruction of posterior cervical ligament complex and so on. Compared with the previous posterior cervical laminectomy, posterior double open-door cervical surgery can achieve the same effect of spinal cord decompression, and can effectively preserve the posterior structure and movement segments of cervical spine, and reduce the incidence of postoperative complications. And it is not easy to close the door again.

Round 2

Reviewer 1 Report

Thank you for the correction to my suggestions. I have read about your response to the peer reviewers.

<About the measurement method>

1. if you measured the C2-7 angle and T1Slope using only the standing X-ray image, then I don't think you should use the CT image as shown in Figure 4. (Why not remove Figure 4? This is a cause for doubt as in question 2.)

2. If you were unable to determine this on X-ray images and sometimes used the C6 endplate or C7 Slope as a substitute or used CT image measurements, you should describe it as such.

<New description of the discussion section>

1. P.7-8/11 "At present, …after cervical laminoplasty.”, You cite references to cervical posterior fusion surgery, which is not the subject of this study, Is this an essential discussion? (I am uncomfortable with that. )

Author Response

Comments and Suggestions for Authors

Thank you for the correction to my suggestions. I have read about your response to the peer reviewers.

<About the measurement method>

  1. if you measured the C2-7 angle and T1Slope using only the standing X-ray image, then I don't think you should use the CT image as shown in Figure 4. (Why not remove Figure 4? This is a cause for doubt as in question 2.)

Answer: You are right. We replaced the initial figures with the image (information) of another patient and both figures are X-ray images. We revised the manuscript. We inserted the two figures and caption in the manuscript.

Male, 62 years old, preoperative diagnosis: CSM, cervical vertebra 3-7 posterior double door. The preoperative tilt angle of T1 was 35.27°, the preoperative cervical curvature was 34.99°, the postoperative follow-up was 3 months, the cervical curvature was 28.07°, and the cervical curvature lost 6.92°.

  1. If you were unable to determine this on X-ray images and sometimes used the C6 endplate or C7 Slope as a substitute or used CT image measurements, you should describe it as such.

Answer: We rechecked all the x-rays.

<New description of the discussion section>

  1. 7-8/11 "At present, …after cervical laminoplasty.”, You cite references to cervical posterior fusion surgery, which is not the subject of this study, Is this an essential discussion? (I am uncomfortable with that. )

Answer:  You are correct. We inserted the two references and discussed their findings at the request and suggestion of another respected reviewer. Now we have deleted the sentences as they are not much related to the topic of the study.

Reviewer 2 Report

Authors have sufficiently responded to reviewer remarks. 

Acceptable.

Author Response

Comments and Suggestions for Authors: Authors have sufficiently responded to reviewer remarks. 

Comments on the Quality of English Language: Acceptable.

Answer: we appreciate the time and efforts of the respected reviewer. We revised the whole manuscript to remove language issues. We also asked an expert to copy-edit and revise the manuscript. We removed two references at the request of reviewers as they are not related to the topic of the study. We also revise introduction, methods and discussion to improve the transparency and presentation of the study. We hope the revised manuscript meet the journal standard to be accepted for publishing.

Submission Date: 03 May 2023

Date of this review: 21 Jul 2023 13:36:30

Reviewer 3 Report

The authors revised well according to my requests.  This 2nd paper is refined better.  I am sure this revised manuscript is ready for publication.  For these reasons, I judged this 2nd manuscript is appropriate for brain sciences journal.

Author Response

Answer to - Reviewer-3

Comments and Suggestions for Authors

The authors revised well according to my requests.  This 2nd paper is refined better.  I am sure this revised manuscript is ready for publication.  For these reasons, I judged this 2nd manuscript is appropriate for brain sciences journal.

Answer: we appreciate the time and efforts of the respected reviewer. We revised the whole manuscript to remove language issues. We also asked an expert copy-editor to revise the manuscript. We also removed two references at the request of reviewers as they are not related to the topic of the study. We hope the revised manuscript meet the journal standard to be accepted for publishing.

Submission Date

03 May 2023

Date of this review

21 Jul 2023 10:03:43
